# Predictive Microbial Markers Distinguish Responders and Non-Responders to Adalimumab: A Step Toward Precision Medicine in Ulcerative Colitis

**DOI:** 10.3390/microorganisms13081941

**Published:** 2025-08-20

**Authors:** Shaghayegh Baradaran Ghavami, Arfa Moshiri, Carola Bonaretti, Maryam Farmani, Margherita Squillario, Eddi Di Marco, Shabnam Shahrokh, Hedieh Balaii, Maria Valeria Corrias, Mirco Ponzoni, Amir Sadeghi, Roberto Biassoni

**Affiliations:** 1IBD Department, Basic and Molecular Epidemiology of Gastrointestinal Disorders Research Center, Research Institute for Gastroenterology and Liver Diseases, Shahid Beheshti University of Medical Sciences, Tehran 1985717413, Iran; shaghayegh.bghavami@gmail.com (S.B.G.); mfarmani171@gmail.com (M.F.); shabnamshahrokh@gmail.com (S.S.); hedie.baalaei@gmail.com (H.B.); amirsadeghimd@yahoo.com (A.S.); 2Experimental Therapies in Oncology, IRCCS Istituto Giannina Gaslini, 16147 Genova, Italy; arfa.moshiri@nih.gov (A.M.); mariavaleriacorrias@gaslini.org (M.V.C.); 3Molecular Diagnostics, IRCCS Istituto Giannina Gaslini, 16147 Genova, Italy; carolabonaretti@yahoo.it (C.B.); robertobiassoni@gmail.com (R.B.); 4IRCCS Ospedale Policlinico San Martino, 16131 Genova, Italy; margherita.squillario@hsanmartino.it; 5Central Laboratory, IRCCS Istituto Giannina Gaslini, 16147 Genova, Italy; eddidimarco@gaslini.org

**Keywords:** ulcerative colitis, stool metagenomics, mucosal metagenomics, Mayo score, TNF alpha inhibitors therapy

## Abstract

Ulcerative colitis (UC) is a chronic, relapsing inflammatory disease of the colon, often associated with gut microbial dysbiosis. Although anti-TNF-α agents, such as Adalimumab (Cinnora^®^), are used to treat moderate-to-severe UC, the treatment response is highly variable. Identifying early microbial biomarkers of response could help support personalized therapeutic strategies and prevent unnecessary exposure to ineffective treatments. However, the long-term effects of anti-TNF therapy on both stool and mucosal microbiota remain poorly understood. This prospective longitudinal study included 23 corticosteroid-refractory or -dependent UC patients who started Adalimumab after endoscopy-confirmed flare-ups. Stool samples and inflamed colonic biopsies were collected at baseline, and 3 and 6 months. Microbiota profiling was performed using 16S rRNA sequencing. Microbial changes were analyzed over time and compared between responders (Mayo score 0–1) and non-responders (Mayo score ≥ 2). Sixty percent of patients achieved clinical remission. In responders, stool microbiota showed increased *Bacteroidetes* and decreased *Proteobacteria* abundances, along with an enrichment of beneficial taxa including *Faecalibacterium prausnitzii*, *Bifidobacterium*, and *Akkermansia muciniphila*. Mucosal microbiota exhibited persistent dysbiosis, characterized by an increase in *Proteobacteria* and a reduced *Firmicutes*/*Proteobacteria* ratio. Notably, responders showed distinct compartment-specific microbial changes, with a decrease in *Gammaproteobacteria* in stool and an increase in *Corynebacterium* in tissue. Adalimumab induces divergent microbial changes in stool and mucosa. While stool microbiota trends toward eubiosis in responders, persistent mucosal dysbiosis may reflect asymptomatic inflammation. These findings underscore the importance of niche-specific microbiome profiling in UC and support its integration into personalized treatment monitoring.

## 1. Introduction

Inflammatory bowel disease (IBD) is a condition characterized by inflammation of the intestinal wall, with periods of relapse triggered by an imbalance in the immune system and gut microbiota [1]. There are two main types of IBD: ulcerative colitis (UC), which is confined to the colon, and Crohn’s disease (CD), which can affect any part of the gastrointestinal (GI) tract, from the mouth to the peri-anal area. The primary symptoms of IBD include bloody diarrhea, abdominal pain, weight loss, fatigue, and fever [2].

The exact cause of IBD remains unknown, though it is believed to result from an overactive immune response triggered by a compromised mucosal barrier, which is influenced by intestinal microbiota dysbiosis and environmental factors in genetically predisposed individuals [3,4]. The prevalence of IBD is increasing globally, becoming a significant public health concern in both developed and developing countries. Projections suggest that the IBD burden may reach 30 million patients by 2035 [5,6]. Despite ongoing research into its underlying causes, a cure for IBD remains elusive, with treatment primarily focused on long-term disease management. Recent studies emphasize that the main therapeutic goal should be to manage the disease effectively and achieve sustained remission [7].

Treatment strategies for IBD involve a variety of medications tailored to the patient’s disease stage. Anti-TNF (tumor necrosis factor) agents are biologics often used as first-line therapies, either alone or in combination with other drugs, to treat active moderate-to-severe UC and CD [7,8]. These agents are particularly recommended for patients who do not respond to traditional therapies. Although anti-TNF therapy is generally effective, approximately 33% of patients are primary non-responders [9]. The advent of biologic therapies has significantly reduced surgical rates and hospitalizations, improving patients’ quality of life. However, understanding the causes of primary and secondary non-responsiveness remains crucial.

The gut microbiome plays a critical role in IBD [10], influencing physiological functions such as intestinal integrity, energy metabolism, pathogen defense, and immune regulation. Recent studies have demonstrated that specific microbiota profiles can impact the success of biological therapies, particularly anti-TNF treatments [11,12]. The microbiota composition can influence immunotherapy outcomes by producing metabolites that help regulate the colon environment, oxygen levels, and the balance of harmful gut bacteria. Consequently, microbiota composition can significantly affect treatment responses and disease recurrence [13].

In terms of diagnosis, the most sensitive marker of intestinal inflammation in IBD, aside from endoscopy, is fecal calprotectin, which strongly correlates with endoscopic disease activity [14]. Since gut microbiota composition can vary based on factors like the home microbial environment and diet, this study aims to investigate microbiome differences between responders and non-responders to anti-TNF therapy in UC patients. Additionally, the study seeks to develop a practical microbial panel using a precision medicine approach to optimize treatment selection for each individual patient.

From a diagnostic perspective, fecal calprotectin has become the most sensitive non-invasive biomarker of intestinal inflammation, showing strong alignment with endoscopic indices of disease activity in IBD [14]. Considering that microbiome composition can vary based on geography, environment, and diet, this study aims to explore differences in the gut microbiota of UC patients who respond versus those who do not respond to anti-TNF therapy. Moreover, the study aims to identify a clinically relevant microbial signature to help predict treatment outcomes.

## 2. Materials and Methods

### 2.1. Patients

We analyzed the microbiomes of stool samples and intestinal biopsies from the inflamed mucosa of 23 patients (mean age 32 ± 10 years at the time of sampling), including 12 females, all of whom had severe ulcerative colitis (UC) (endoscopic Mayo score = 3) [15]. The duration of disease in these patients was more than five years (mean 5.3 ± 4.4 years). Among these patients, 14 had pancolitis, while 9 had left-sided colitis (see Table 1 for details). The clinical history of these patients indicated corticosteroid dependency or refractoriness, which led to their selection for treatment with Adalimumab (Cinnora^®^), a tumor necrosis factor (TNF)-alpha inhibitor approved by the FDA for treating ulcerative colitis following endoscopic confirmation of a flare-up. At the start of therapy, more than 16 out of the 23 patients had severe symptoms. Adalimumab was administered every two weeks, with an initial dose of 160 mg, followed by 80 mg for the second dose, and then 40 mg every two weeks for a treatment period of 3 to 6 months. Patients were evaluated at baseline and after 3 or 6 months of Adalimumab treatment through clinical, biochemical, and endoscopic assessments. In detail, the endoscopic procedure involved collecting both stool samples and sections of inflamed mucosa without any bowel cleansing for the first two longitudinal samples. For the samples collected after six months of therapy, standard bowel cleansing was performed. All specimens were stored at −70 °C until DNA extraction. The study was approved by the Research Institute for Gastroenterology & Liver Diseases on 10 June 2019, with approval ID (IR.SBMU.RCGLD.REC.1398.018) issued to Dr. Shaghayegh Baradaran Ghavami. All experiments were conducted under the Helsinki Declaration.

### 2.2. Microbiome Analysis

DNA was extracted from stool and tissue samples using the QIAamp Fast DNA Stool Mini Kit or DNeasy Blood & Tissue Kit (Qiagen, Hilden, Germany). The eluted DNA was analyzed for both quality and quantity through spectrophotometric and fluorometric methods. DNA concentration was determined using the fluorometric quantification assay (Qubit by ThermoFisher Scientific, Waltham, MA, USA). A total of 3 ng of extracted DNA was used to amplify the 16S ribosomal RNA gene using the Ion 16S™ Metagenomics Kit (ThermoFisher Scientific). This method enables PCR amplification of 7 out of 9 informative 16S polymorphic regions (set-1: V2, V4, V8; set-2: V3, V6–7, V9), according to the manufacturer’s protocol [16].

Library preparation was performed using the IonPlus Library Kit for AB Library Builder (Thermo Fisher Scientific). Up to 16 differently barcoded libraries were then processed using the Ion Chef System, loaded onto an Ion 520 chip, and sequenced on the GeneStudio S5 system (Thermo Fisher Scientific). Data analysis was carried out using the Ion Reporter™ (v 5.18.0.2) suite, with the curated Greengenes (v13.5) and premium curated MicroSEQ ID 16S rRNA reference libraries (v2013.1) databases, applying standard parameters.

### 2.3. Bioinformatics and Statistics

Statistical analysis, visualization, and meta-analysis of microbiome data were performed using the MicrobiomeAnalyst 2 web tool [17,18,19]. Data filtering was applied to remove low-abundance operational taxonomic units (OTUs) (prevalence in 20% of samples) and low-variance OTUs (based on the interquartile range of 10%). Relative abundance comparisons were made using different algorithms, including metagenomeSeq, EdgeR, DESeq2, linear discriminant analysis (LDA), and effect size (LEfSe). The “metagenomeSeq” algorithm uses the zero-inflated Gaussian fit method, which adjusts for imbalanced class distributions (under-sampling) and sparsity in datasets. RNASeq analysis was performed following the EdgeR or DESeq2 differential abundance algorithms.

Supervised analyses were conducted using sparse partial least squares discriminant analysis (sPLS-DA) and random forest learning algorithms [20,21]. The Benjamini–Hochberg false discovery rate (FDR) correction was applied to *p*-values for multiple comparisons (FDR < 0.05), unless otherwise specified. The q2-longitudinal (q2-l) plugin, which supports statistical and visual comparisons in longitudinal study designs and paired samples, was used to assess how samples changed between different observational “states”. “States” were typically related to time or environmental gradients, and for paired analyses, sample pairs represented the same individual at two different time points. In methodological comparisons, paired samples were from the same individual at the same time but processed with different methods [22].

The q2-sample classifier (q2-sa) was used to predict sample metadata. This supervised learning approach allows for prediction of metadata (e.g., treatment outcomes) based on microbiota composition. Predicted targets included discrete sample classes (for classification problems) or continuous values (for regression problems). Feature volatility analysis (F) was used to identify microbiome features predictive of metadata columns (e.g., time), and relative frequencies of these features across states were visualized with interactive plots. A supervised learning regressor was used to identify key features and assess their ability to predict sample states.

We employed the random forest algorithm, an ensemble machine learning method designed for classification and regression tasks [21]. It utilizes multiple decision trees, reducing the risk of overfitting often associated with individual trees. The random forest algorithm also provides feature importance estimates, helping to identify which features most contribute to the model’s predictions. One metric used to assess feature importance is the mean decrease accuracy, which quantifies the impact of permuting individual feature values on the model’s overall accuracy. Features with a significant impact on model performance show a high decrease in accuracy when permuted, highlighting their relevance.

Weighted correlation network analysis (WGCNA) was applied to explore relationships between taxa and clinical parameters [23]. This method identifies clusters of taxa that exhibit strong correlations across different samples or conditions. By grouping taxa with similar behavior into modules or clusters, we measured the strength of their associations using a weighted correlation network. The clinical variables included in the analysis were:

Age: continuous variable

Sex: Male (1), Female (0)

Age at diagnosis: continuous variable

Disease duration in years: continuous variable

Disease extent prior to therapy: Left-Sided Colitis (1), Pancolitis (0)

Disease severity before therapy: Severe (1), Mild to Moderate (2), Moderate (3), Moderate to Severe (4)

Endoscopic activity of the rectum after 3 months of Cinnora^®^ therapy: Remission (0), Flare-up (1)

Disease state 6 months after Cinnora^®^ therapy: Remission (0), Flare-up (1)

Corticosteroid use: Dependency (1) or Refractoriness (0)

Azathioprine use: Dependency (1) or None (0)

Comorbidities (e.g., primary sclerosing cholangitis, rheumatoid arthritis, pyoderma gangrenosum): Yes (1), No (0)

Alcohol consumption: Yes (1), No (0)

Smoking habits: Yes (1), No (0)

Platelet count: continuous variable

White blood cell (WBC) count: continuous variable

Hemoglobin (Hb): continuous variable

## 3. Results

### 3.1. Longitudinal Analysis of Treatment with the Cinnora^®^ (TNF)—Alpha Inhibitor

A total of 23 patients, all suffering from ulcerative colitis (UC) for 5.3 ± 4.4 years, were included in the study. Following an endoscopy that detected a flare-up of UC, they initiated treatment with the biologic agent Cinnora^®^ (Adalimumab), as these patients were either corticosteroid-refractory or -dependent. Clinical remission was achieved in 60% of the patients. Remission was assessed using the four grades of the Mayo endoscopic score (0: no inflammation; 1: mild; 2: moderate; and 3: severe) [15]. Of the 23 patients, 8 achieved complete remission (Mayo = 0), while an additional 5 experienced symptom improvement, with only mild inflammation (Mayo = 1) (Table 1).

### 3.2. Microbiome Visual Analysis

The analysis of 59 stool samples yielded 223,074 ± 168,263 (median 190,361) 16S-mapped reads, containing 1830 ± 1312 operational taxonomic units (OTUs). In contrast, the 45 inflamed mucosa samples provided 230,544 ± 164,503 (median 220,431) 16S-mapped reads and 2536 ± 1584 OTUs. All the samples in the study were collected from patients with active disease, as confirmed by endoscopy.

The baseline microbiome composition of stool samples from the 23 UC patients showed *Firmicutes* as the most abundant phylum, with a median relative abundance of 41.1%, followed by *Bacteroidetes* (31.0%) and *Proteobacteria* (13.2%). In contrast, microbial samples from the inflamed mucosa before treatment showed similar median abundances for *Firmicutes* (28.4%) and *Bacteroidetes* (29.5%), while *Proteobacteria* had a median value of 14.1% (Appendix A).

For the longitudinal analysis, we evaluated changes in the microbiome during treatment with Adalimumab. Stool samples showed a slight increase in the abundance of *Bacteroidetes*, with median values increasing from 31.0% before treatment (pre) to 38.6% at 3 months (R2), and to 40.5% at 6 months (R3). A slight decrease in the abundance of *Firmicutes* and *Proteobacteria* was also observed. Specifically, *Firmicutes* showed median values of 41.1% (pre), 40.9% (R2), and 38.4% (R3), while *Proteobacteria* showed median values of 13.2%, 11.1%, and 10.9%, respectively.

Analysis of the microbiome from inflamed mucosal biopsies revealed more pronounced shifts. The abundance of *Firmicutes* decreased from a median of 28.4% at baseline to 23.5% after 6 months of treatment. In contrast, *Proteobacteria* showed an opposite trend, increasing from a median of 14.1% to 34.5%. This resulted in a decrease in the *Firmicutes/Proteobacteria* (F/P) ratio from 2.01 before treatment to 0.68 after 6 months of anti-TNF therapy, reflecting significant dysbiosis (Appendix A).

### 3.3. Alpha and Beta Diversity Analysis

In the analysis of alpha diversity indices, both community richness (Chao1 index) and the richness and evenness (Shannon and Simpson indices) were considered. Although no statistical significance was observed, the analysis provided valuable insights into the two ecological niches (stools and inflamed intestinal mucosa) across samples before treatment (pre) and within the non-responsive (NR) and responsive (R) groups. Notably, differences in behavior were observed between stool and tissue-derived samples.

The Chao1 richness index indicated a decreasing trend in stool microbiomes; however, no such trend was observed in the tissue biopsies. Of particular interest, the stool microbiome of the responders showed a slight increase in both the Shannon index (median: 1.74 [pre], 1.96 [R]) and the Simpson index (median: 0.70 [pre], 0.81 [R]). In contrast, mucosal biopsy samples showed a slight decrease in the same indices: Shannon (2.3 vs. 2.1) and Simpson (0.83 vs. 0.81) (Appendix A).

Beta diversity profiling further illustrated that microbial populations in stool and inflamed tissue samples were distinct, as shown by the Bray–Curtis index. PERMANOVA analysis revealed no statistically significant differences among pre-treated, responsive, and non-responsive samples treated with the TNF alpha inhibitor Cinnora^®^, either collectively or in pairwise comparisons (Appendix A).

### 3.4. Extension of Pathology

The stool microbiome of pancolitis patients exhibited a high abundance of *Bacteroidetes* (median 42.5%), followed by *Firmicutes* (median 30.7%). In contrast, the microbiomes of left-sided colitis patients were enriched in *Proteobacteria* (median 42.4%). The microbiome data were then analyzed using q2-longitudinal, a bioinformatics tool designed for paired and longitudinal data analysis [22]. Strong positive correlations were identified between stool samples and disease extension, particularly concerning the relative abundance of several taxa: *Prevotellaceae*, *Lachnospiraceae*, *Lactobacillaceae*, *Enterobacteriaceae*, *Clostridiaceae*, *Bifidobacteriaceae*, *Streptococcaceae*, *Veillonellaceae*, *Pasteurellaceae*, *Burkholderiaceae*, and pancolitis. Similarly, a strong positive correlation was found between *Bacteroidaceae*, *Hyphomicrobiaceae*, *unclassified Clostridiales*, and certain *Brucellaceae* species in patients with left-sided colitis (Appendix A).

Additionally, the greatest increases in microbial abundance were observed for *Pseudomonas aeruginosa*, *Dialister succinatiphilus*, *Ruminococcus callidus*, *Rothia dentocariosa*, and *Clostridium aldenense* in inflamed biopsies from pancolitis patients. In contrast, the family *Cytophagaceae*, along with genera *Escherichia* and *Cronobacter*, and the species *Abiothophia defectiva*, were particularly abundant in patients with left-sided colitis (Appendix A).

### 3.5. Differential Abundance Analysis of Patients Showing Remission Compared to Those Maintaining an Inflammatory Status over Therapy

The microbiome analysis revealed marked differences in the abundance of *Proteobacteria* between stool and tissue biopsies. For stool samples, there was a noticeable trend toward a decline in *Proteobacteria* abundance, with a median value of 13.2% before treatment, which decreased to 11.1% in responders and 9.4% in non-responders. In contrast, tissue biopsy samples from responders exhibited an increase in the abundance of this phylum, from 14.1% to 22.4%, while non-responders had a median of 17.4%. *Bacteroidetes* also showed differential changes, with an increase in abundance from a median of 29.5% pre-therapy to 44.9% in non-responders and 22.0% in responders (Figure 1).

A deeper analysis of the microbiome in anti-TNF-α therapy responders revealed a notable abundance of beneficial genera such as *Bifidobacterium*, *Bacteroides*, *Alistipes*, *Christensenellaceae*, *Peptococcaceae*, *Lactobacillus*, and *Phascolarctobacterium*, which were previously underrepresented. This increase in beneficial bacteria was associated with a reduction in the endoscopic Mayo score, indicating improvement. Conversely, there was a significant increase in *Corynebacterium* and *Aeromonas* in the mucosal biopsies of patients who responded to biologics therapy (Appendix A).

Additionally, differential abundance analysis of stool microbiomes demonstrated that responders had lower *Proteobacteria* compared to non-responders (log_2_ fold change of −2.7925, FDR = 0.003). Responders showed an even greater decline in *Proteobacteria* abundance after treatment compared to pre-treatment levels (log_2_ fold change of −3.3756, FDR 1.3 × 10^−5^), with both comparisons being statistically significant at different time points (Figure 2a). On the other hand, *Actinobacteria* levels were slightly higher in responders than in non-responders (log_2_ fold change of 2.2252, FDR = 0.041) (Figure 2b).

Stool samples from UC patients undergoing anti-TNF alpha therapy showed that several microorganisms, including *Lactobacillus gasseri*, *Staphylococcaceae*, *Corynebacteriaceae*, and *Comamonadaceae*, showed a reduction in abundance post-therapy. In contrast, several *Firmicutes* descendants, such as *Clostridium disporicum*, *Roseburia* sp., and *Turicibacter sanguinis*, *Bacteroides ovatus*, *Odoribacter splanchnicus*, increased in abundance, surpassing pre-therapy levels after three and six months of treatment (Figure 3a,b).

More of the same data demonstrated a dramatic reduction in the abundance of *Gammaproteobacteria* and several descendant taxa after six months (Figure 3b).

We further conducted a differential abundance analysis by categorizing samples according to the Mayo endoscopic score. All comparisons were made against samples collected before treatment (Mayo score = 3). Samples from patients classified as responders, with Mayo scores of “0” or “1” (indicating no or only minor inflammation), demonstrated significant microbiome changes after 3 or 6 months on TNF inhibitors. Notably, half of the genera with increased abundance in responders belonged to the *Firmicutes* phylum. Among all the genera analyzed using the EdgeR method, *Phascolarctobacterium* showed the most significant increase in abundance in the responder group, a pattern not observed in non-responders (Mayo scores “2” or “3”). Furthermore, a significant decrease was noted in the abundance of several genera within the class *Gammaproteobacteria*, such as *Escherichia*, *Escherichia/Shigella*, *Cronobacter*, and *Trabulsiella* (Appendix A).

In mucosal biopsies, tissue samples from responders (Mayo scores “0” and “1”) revealed an increase in abundance of certain *Proteobacteria*, including *Aeromonas*, *Trabulsiella*, *Salmonella*, and *Eggerthella* (a member of the *Actinobacteria* phylum). Conversely, there was a noticeable scarcity of *Firmicutes* genera such as *Anaerococcus*, *Finegoldia*, *Peptoniphilus*, and *Streptococcus*, as well as some *Bacteroidetes*, such as *Cloacibacterium* and *Butyricimonas* (Appendix A).

Analysis of the stool microbiome further revealed distinct microbial profiles when comparing patients in remission (R) to those in flare-up (F). Anti-inflammatory strains, such as *Faecalibacterium prausnitzii*, *Akkermansia muciniphila*, *Phascolarctobacterium faecium*, *Clostridium hiranonis*, and *Eubacterium hallii*, were more abundant in patients in remission. In contrast, bacteria like *Escherichia coli* and *Cronobacter turicensis* were predominantly found in the flare-up group (Appendix A).

In inflamed tissue samples, the treatment with Cinnora^®^ led to an increase in *Proteobacteria* abundance after 3 and 6 months, particularly in descendants of the *Gammaproteobacteria* and *Alphaproteobacteria* classes. Families like *Enterobacteriaceae* and *Bradyrhizobiaceae* showed increased abundance post-treatment (Figure 4). In contrast, *Firmicutes*, including *Lactobacillales* and *Clostridiales*, were dominant in tissue samples before treatment, but their presence gradually decreased throughout therapy (Figure 4).

### 3.6. Disease State

We performed a supervised learning analysis using the q2-sample-classifier (q2-sa) to assess microbial frequencies associated with either remission or inflammation (flare-ups), aiming to predict related sample metadata.

After six months of treatment, stool samples from patients in remission showed a higher frequency of genera associated with positive outcomes, including *Lactobacillus*, *Prevotella*, *Faecalibacterium*, *Clostridium*, *Dorea*, and *Turicibacter*. These genera were significantly more abundant in patients experiencing ulcerative colitis (UC) remission.

Conversely, inflammation-related samples from patients in flare-ups demonstrated an increased presence of genera such as *Blautia*, *Parasutterella*, *Sutterella*, *Cronobacter*, and *Granulicatella* (Figure 5).

We also examined the microbiome in patients categorized based on their response to corticosteroid treatment (corticosteroid dependency vs. resistance). Stool samples from patients dependent on corticosteroids exhibited a high abundance of genera like *Phascolarctobacterium*, *Escherichia/Shigella*, and *Kocuria*, suggesting a possible link to corticosteroid dependency. In contrast, patients resistant to corticosteroid therapy had higher levels of *Lactobacillus*, *Streptococcus*, *Bifidobacterium*, *Gemminger*, *Granulicatella*, and *Actinomyces*, which were associated with corticosteroid resistance (Figure 6).

### 3.7. Supervised Random—Forest Analysis

We employed a machine learning approach using a random forest classifier to identify microorganisms linked to either responsiveness (R) or non-responsiveness (NR) to anti-TNF alpha therapy, as well as those present in samples taken prior to treatment.

The stool microbiome of patients who responded to the therapy was notably enriched in microorganisms from the *Firmicutes* phylum, such as *Gemminger*, *Erysipelotrichia*, *Phascolarctobacterium faecium*, and various *Clostridia* species. In contrast, non-responders exhibited microbiomes predominantly dominated by *Proteobacteria*.

Interestingly, in the inflamed biopsy tissues of patients receiving Cinnora^®^ (Adalimumab), a predominant presence of *Gammaproteobacteria* was consistently observed, particularly *Escherichia coli* and *Trabulsiella odontotermitis*. Among patients whose endoscopic Mayo score improved, indicating they were responders, the *Proteobacteria* identified were exclusively from the *Gammaproteobacteria* class. This finding suggests a potential association between the presence of these microorganisms in tissue biopsies and therapeutic outcomes, emphasizing the complex relationship between mucosal microbiota and inflammatory conditions (Appendix A).

For non-responders, stool microbiomes were notably enriched in *Gammaproteobacteria* taxa, while their mucosal biopsies showed high abundance of *Deltaproteobacteria* and *Escherichia* (Appendix A).

### 3.8. Weighted Correlation Network Analysis (WGCNA)

To explore the relationships among different variables, including microbiome features and clinical data, we applied weighted correlation network analysis (WGCNA). The complete list of identified clusters and their associations can be found in the figure legend, which details the groups of variables that showed strong correlations. These clusters were statistically analyzed for significance, with *p*-values adjusted using the false discovery rate (FDR) method.

In the stool microbiome of UC patients who did not respond to anti-TNF alpha treatment, we observed a strong positive correlation between the disease state after three months of therapy, comorbidities, and white blood cell (WBC) counts. Remarkably, more than 70% of the most prevalent microbial species in these samples belonged to the *Firmicutes* phylum (Appendix A).

Conversely, in inflamed tissue biopsies from non-responders, the microbiome was predominantly composed of *Proteobacteria*, with over 50% of the species belonging to this phylum, and 73% from the *Gammaproteobacteria* class. This group included species such as *Cronobacter malonaticus*, *Cronobacter turicensis*, *Enhydrobacter aerosaccus*, *Enterobacter asburiae*, *Escherichia coli*, *Escherichia vulneris*, *Leclercia adecarboxylata*, *Pantoea agglomerans*, *Salmonella enterica*, and *Trabulsiella odontotermitis* (Appendix A).

In contrast, in the stool microbiome of patients who responded to treatment, more than half of the microorganisms belonged to *Firmicutes*, and these showed a positive correlation with disease state after three months and disease severity (Appendix A). In inflamed tissue biopsies, the presence of *Proteobacteria* was positively associated with a specific cluster of clinical variables, and after three months of treatment, there was a strong positive correlation between disease state and the presence of *Proteobacteria* in more than 85% of the studied taxa (Appendix A).

### 3.9. Inferring Functional (Metabolic) Pathways Characterizing UC Patients Under Adalimumab Therapy

To explore the potential functional alterations in the gut microbiome of UC patients undergoing Adalimumab therapy, predicted metabolic pathways were inferred using PICRUSt2 (Phylogenetic Investigation of Communities by Reconstruction of Unobserved States 2). This approach enabled the estimation of microbial gene content from 16S rRNA gene sequencing data based on known databases such as KEGG and MetaCyc [24,25,26]. Previously published data in newly diagnosed type 1 diabetes patients showed that a good correlation existed between the pathways found with a proteomic approach and the ones inferred with PICRUSt based on 16S sequencing; indeed, 8 of the 13 pathways previously identified through a direct metaproteomics approach [27] were also detected among the dysregulated pathways predicted by the PICRUSt computational pipeline using our set of data [28].

By comparing microbiome data from UC patients before treatment and after 3 or 6 months of therapy, researchers identified several statistically significant functional pathways. As therapy progressed, the number of significant KEGG pathways increased, reflecting a shift in microbial metabolic activity in both inflamed tissues and fecal microbiomes. Notably, after six months of Adalimumab treatment (R3), significant changes in metabolic pathways were detected in both stool samples and mucosal biopsies. Specifically, 299 features were identified as significant in stool samples, while 910 pathways were enriched in mucosal biopsy microbiomes. Among these, 48 pathways in stool microbiomes and 883 in mucosal biopsies exhibited increased abundance after TNF alpha inhibitor therapy (Appendix A).

### 3.10. Key Pathways Enriched in Mucosal Biopsies but Not Stool Microbiomes

A closer inspection revealed that several pathways increased in inflamed tissue microbiomes but did not show similar enrichment in stool microbiomes. These included pathways linked to the two-component system and biofilm formation, both of which play critical roles in microbial adaptation and virulence.

Type 1 Fimbriae Pathways:

Ten pathways related to type 1 fimbriae were notably enriched in mucosal tissue microbiomes. These pathways include genes involved in fimbrial assembly, transport, and regulation:

fimD (outer membrane usher protein, K07347)

fimB (type 1 fimbriae regulatory protein, K07357)

fimC (fimbrial chaperone protein, K07346)

fimG (minor fimbrial subunit, K07349)

fimA (major type I fimbrial subunit, K07345)

fimH (minor fimbrial subunit, K07350)

fimF (minor fimbrial subunit, K07348)

fimI (fimbrial protein, K07351)

These genes encode proteins that are involved in microbial adhesion, biofilm formation, host interactions, and virulence [29].

Curli Fibers Pathways:

Similarly, several components of curli fibers were significantly enriched in mucosal biopsies. These include genes involved in curli fiber assembly and regulation:

csgA (major curlin subunit, K04334)

csgC (curli production regulator, K04336)

csgB (minor curlin subunit, K04335)

csgE (curli production assembly/transport component, K04337)

csgD (Lux-family transcriptional regulator, K04333)

csgG (curli production assembly/transport component, K06214)

csgF (curli production assembly/transport component, K04338)

These genes contribute to surface attachment, biofilm formation, microbial persistence, and host-pathogen interactions, which are essential for microbial colonization and virulence [30]. Using PICRUSt2 analysis of 16S sequencing data from stool microbiomes, 347 KEGG-inferred pathways were statistically significant. Among these, 48 pathways showed increased abundance after six months of TNF-α inhibitor treatment. Specifically, seven pathways were related to energy metabolism, six to cell wall/envelope biogenesis, five to membrane transport and drug resistance, four each to carbohydrate/amino acid metabolism, and to stress responses. Finally, three are associated with protein folding and DNA repair, and two with transporters and channels. Among the 251 inferred pathways that showed a decrease in the abundance after 6 months of TNF-α inhibitor administration, the great majority (162) were pathways associated with metabolic pathways, and sugar and amino acid metabolism.

## 4. Discussion

In this six-month study of 23 patients with ulcerative colitis (UC) treated with TNF alpha inhibitor biologics, 60% of patients achieved clinical remission. Eight patients experienced complete remission, with an endoscopic Mayo score of 0, while five showed only mild residual inflammation (Mayo score of 1). Both groups were classified as responders to the treatment.

Although the total number of subjects, as well as the number per group, represents a limitation of this analysis, we believe that our longitudinal and dual-niche design provides meaningful insights, and we are planning to expand the sample size in future follow-up studies.

Before treatment, the stool microbiome was predominantly composed of *Firmicutes*, followed by *Bacteroidetes* and *Proteobacteria*. In contrast, microbial communities in inflamed mucosal biopsies demonstrated a more balanced composition of *Firmicutes* and *Bacteroidetes*, with *Proteobacteria* accounting for about half the abundance of the other two phyla.

After six months of treatment, stool samples from responders showed an increase in *Bacteroidetes* abundance and a decrease in *Proteobacteria*, a trend opposite to the one observed in mucosal biopsy microbiomes. Here, the proportion of *Proteobacteria* significantly increased, particularly within the *Gammaproteobacteria* class, while *Firmicutes* showed a slight decrease.

These findings suggest that TNF alpha inhibitor therapy significantly alters the stool microbiome, reducing the presence of pro-inflammatory *Proteobacteria*, but this effect is not mirrored in the inflamed mucosal microbiome. The observed increase in *Proteobacteria* in tissue samples is in line with previous studies on UC patients [31] and suggests a link between these microbial changes and UC’s clinical symptoms. Interestingly, *Proteobacteria* showed distinct behavior between stool and tissue samples, with a more pronounced presence in inflamed biopsies post-treatment.

Further analysis using beta-diversity measures highlighted differences between stool and mucosal microbiomes, as patients who responded to anti-TNF alpha treatment exhibited higher Shannon and Simpson diversity indices in stool microbiomes. However, mucosal biopsies showed a decrease in alpha-diversity following treatment, which is consistent with previous observations that UC patients typically have lower alpha-diversity compared to healthy controls [32].

When examining disease extension, a clear distinction was found between pancolitis and left-sided colitis. The stool microbiome of responders displayed a marked reduction in Proteobacteria and an increase in beneficial genera such as *Lactobacillus*, *Faecalibacterium*, and *Clostridium*. In contrast, non-responders maintained high levels of *Proteobacteria*, with *Blautia* and *Parasutterella* genera showing increased abundance in inflammatory samples. Notably, the lower levels of *Faecalibacterium prausnitzii* and *Lactobacillus*, which are considered biomarkers of UC, aligned with poorer clinical outcomes. These findings suggest that *Faecalibacterium prausnitzii*, which is known for its anti-inflammatory properties and its role in gut health [30], may serve as a therapeutic biomarker for UC treatment success.

Longitudinal analysis revealed that several taxa linked to anti-inflammatory properties, including *Bacteroides*, *Alistipes*, and *Phascolarctobacterium*, showed increased abundance in stool samples of responders. These genera are involved in the production of short-chain fatty acids (SCFAs), such as propionate, which has anti-inflammatory effects and promotes gut barrier integrity [33]. Notably, *Phascolarctobacterium faecium*, a known SCFA producer, was highly abundant in stool microbiomes of patients responding to TNF alpha inhibitors, supporting the idea that microbiome composition plays a crucial role in treatment efficacy.

Other species, such as *Bacteroides ovatus*, *Odoribacter splanchnicus*, and *Firmicutes strains* (*Roseburia*, *Turicibacter sanguinis*), also showed an increase in abundance after 3 or 6 months of therapy. These microorganisms are associated with anti-inflammatory activities, SCFA production, and immune modulation [34,35,36,37]. The increase in these taxa during biologic therapy suggests a microbiome recovery that may correlate with mucosal healing and clinical improvement in UC patients. SCFAs have been previously shown to play a key role in inflammatory diseases [38], further corroborating the relationship between gut microbial composition and therapeutic response in UC.

In contrast, *Staphylococcaceae*, which were prevalent in inflamed mucosal tissues before treatment, showed a reduction in abundance after six months of TNF alpha inhibitor therapy, potentially indicating decreased intestinal inflammation and improved barrier function [39]. However, the emergence of *Corynebacterium* and *Aeromonas* in the mucosal microbiome samples, particularly in responders, raises concerns. These bacteria, although traditionally not considered beneficial, may contribute to microbial imbalance or reflect incomplete mucosal healing. *Aeromonas*, in particular, is known to produce harmful substances such as aerolysin and enterotoxins, which can damage gut epithelial cells, weaken the intestinal barrier, and promote inflammation [40,41].

The random forest analysis identified significant differences between the stool microbiomes of responders and non-responders. Responders showed an abundance of *Firmicutes* species, such as *Gemmiger* and other classes like *Erysipelotrichia* and *Negativicutes*, while non-responders had a higher proportion of *Proteobacteria*, thus suggesting a clear distinction in microbial patterns associated with treatment outcomes. The mucosal microbiome also revealed an interesting shift, with *Enterobacteriaceae*, typically associated with gut inflammation, showing increased abundance in patients who responded to TNF alpha inhibitors [42]. This dual behavior of the microbiome across stool and mucosal samples may reflect the complex interactions between microbial communities and disease progression in UC.

Overall, our data indicate that the microbiome plays a crucial role in the response to TNF alpha inhibitors in UC patients, with distinct microbial behaviors observed in stool and inflamed tissue samples. While therapy leads to a reduction in pro-inflammatory taxa in the stool, the mucosal microbiome shows a different pattern, with an increase in *Proteobacteria* and pathogenic genera. These findings highlight the need for further studies to better understand the dynamics of the microbiome and its relationship with UC treatment outcomes, especially in the context of mucosal healing and inflammation resolution.

In conclusion, while the stool microbiome provides a promising target for monitoring therapy and disease remission, the mucosal microbiome may require more attention, particularly in terms of microbial adherence and biofilm formation. The increase in type 1 fimbriae and curli fibers observed in inflamed tissue microbiomes, which facilitate bacterial adhesion and colonization, suggests that while TNF alpha inhibitors may alleviate symptoms and promote recovery in stool-associated microbiomes, their effect on mucosal healing is less clear [43]. This points to the possibility that the persistence of harmful bacterial adhesion in the mucosa may contribute to incomplete remission and recurrent inflammation in UC patients, even after extended periods of treatment.

Future studies will need to address these gaps, particularly with larger cohorts, and explore the stability of the responder phenotype after discontinuation of TNF alpha inhibitor therapy. These findings underscore the complexity of UC treatment and suggest that microbial interventions, alongside biologic therapies, could be crucial for long-term disease management.

## Figures and Tables

**Figure 1 microorganisms-13-01941-f001:**
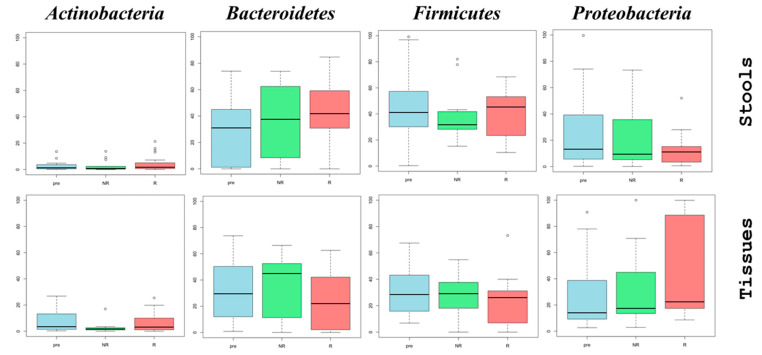
Abundance of the four major phyla in stool microbiomes of UC patients with flare-up inflammation. The abundance of the four major bacterial phyla in the stool microbiomes of ulcerative colitis (UC) patients before and after treatment with anti-TNF alpha (Cinnora^®^) therapy was assessed. Prior to treatment, patients had a flare-up of inflammation, confirmed via endoscopy with a Mayo index of 3 (pre-treatment). Patients were subsequently classified into two groups based on their response to therapy: non-responders (NR), defined by a Mayo index ≥ 2, and responders (R), defined by a Mayo index ≤ 1. The box plot displays the interquartile range (IQR: 25–75%) with the median represented by the line inside the box. Whiskers indicate values within 1.5 times the IQR, while circles outside the whiskers represent outliers. The upper section presents data from stool microbiomes (number of samples: pre-treatment = 23, NR = 13, R = 23), while the lower section displays microbiomes from tissue biopsies (number of samples: pre-treatment = 16, NR = 7, R = 22). The number of patients in each group is indicated in square brackets.

**Figure 2 microorganisms-13-01941-f002:**
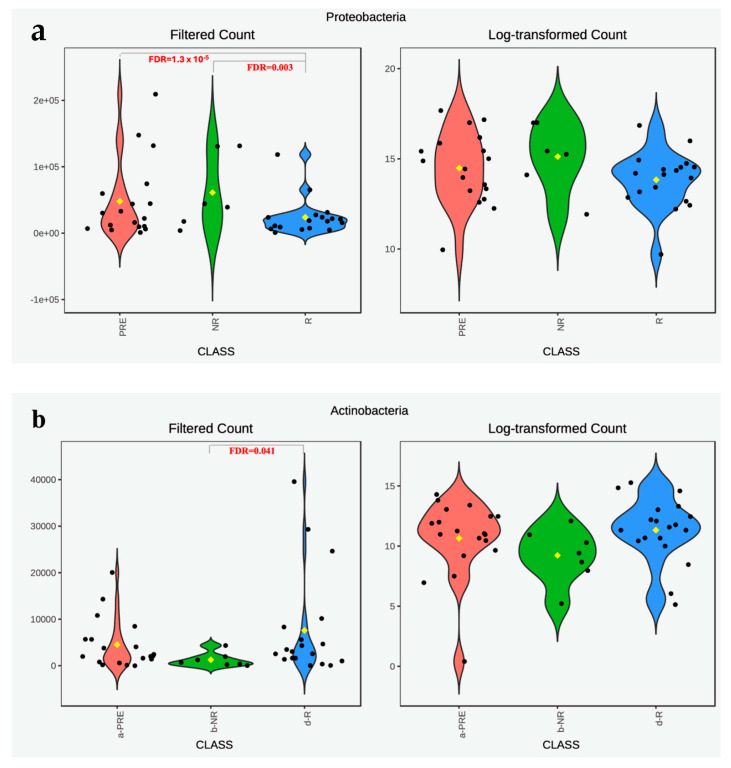
Abundance of *Proteobacteria* (**a**) and *Actinobacteria* (**b**) phyla in stool microbiomes of UC patients. The abundance of *Proteobacteria* (**a**) and *Actinobacteria* (**b**) in the stool microbiomes of ulcerative colitis (UC) patients was assessed before and after treatment with anti-TNF alpha biologics. Patients were diagnosed with a flare-up (pre-treatment), as indicated by a Mayo index of 3. After 3 and 6 months of therapy, patients were classified into non-responders (NR) with a Mayo index ≥ 2, and responders (R) with a Mayo index ≤ 1. Differential abundance analysis was performed using the EdgeR method, with multiple comparison correction applied via the Benjamini and Hochberg false discovery rate (FDR) method.

**Figure 3 microorganisms-13-01941-f003:**
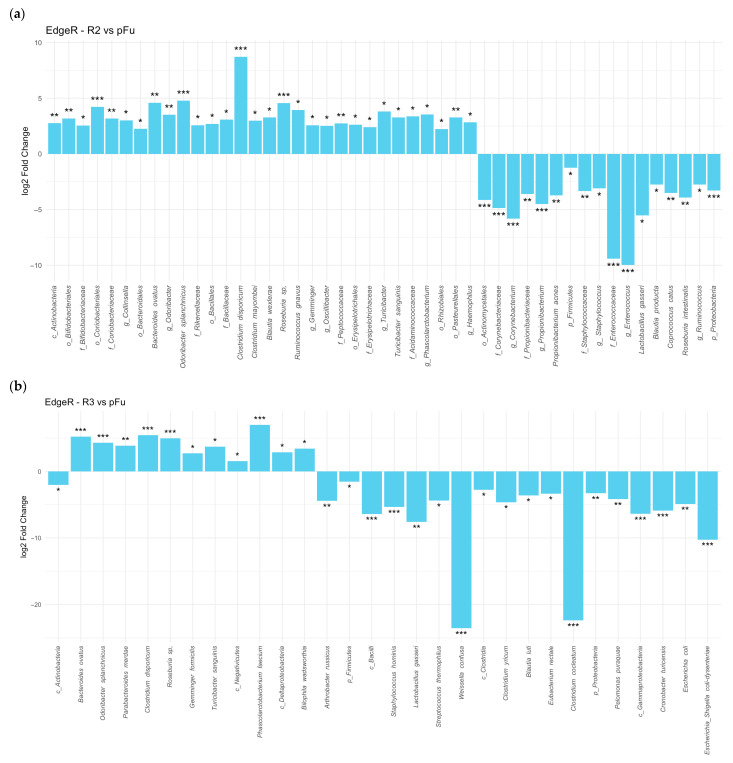
We analyzed the relative abundances of microorganisms in stool microbiome samples from UC patients using the EdgeR algorithm. Post-treatment samples collected after 3 months (R2 [18]) (**a**) and 6 months (R3 [12]) (**b**) of anti-TNF alpha therapy were compared to pre-treatment samples taken during a UC flare-up (pFu [23]). The number of patients in each group is indicated in square brackets. Taxonomic classifications are prefixed as follows: p_ for Phylum, c_ for Class, o_ for Order, f_ for Family, g_ for Genus. To account for multiple comparisons, *p*-values were adjusted using the false discovery rate (FDR), with a threshold of FDR ≤ 0.05 considered statistically significant. Fold-change values are expressed as the base-2 logarithm (log2FC), representing the magnitude and direction of change in microbial abundance between groups. Significance levels are denoted as follows: * FDR ≤ 0.05, ** FDR ≤ 0.01, *** FDR ≤ 0.001.

**Figure 4 microorganisms-13-01941-f004:**
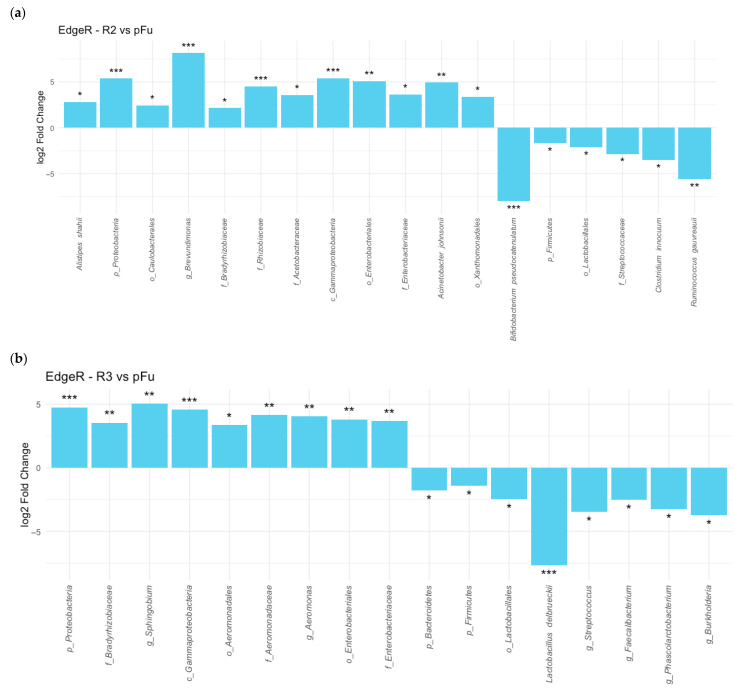
We analyzed the relative abundances of microorganisms in tissue mucosal biopsies and microbiome samples from UC patients using the EdgeR algorithm. Post-treatment samples collected after 3 months (R2 [13]) (**a**) and 6 months (R3 [13]) (**b**) of anti-TNF alpha therapy were compared to pre-treatment samples taken during a UC flare-up (pFu [16]). The number of patients in each group is indicated in square brackets. Taxonomic classifications are prefixed as follows: p_ for Phylum, c_ for Class, o_ for Order, f_ for Family, g_ for Genus. To account for multiple comparisons, *p*-values were adjusted using the false discovery rate (FDR), with a threshold of FDR ≤ 0.05 considered statistically significant. Fold-change values are expressed as the base-2 logarithm (log2FC), representing the magnitude and direction of change in microbial abundance between groups. Significance levels are denoted as follows: * FDR ≤ 0.05, ** FDR ≤ 0.01, *** FDR ≤ 0.001.

**Figure 5 microorganisms-13-01941-f005:**
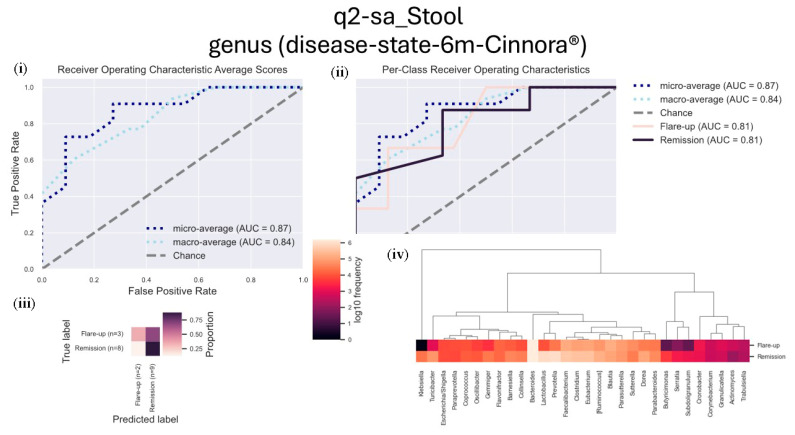
Longitudinal analysis of genus abundance in stool samples from UC patients after six months of Cinnora^®^ treatment. This figure presents the genus abundance in stool samples from patients experiencing flare-ups and those in remission after six months of treatment with Cinnora^®^. The following plots are included: (**i**) the receiver operating characteristic (ROC) curve showing the overall performance of the model, (**ii**) per-class ROC curves for flare-up and remission groups, (**iii**) a confusion matrix comparing true labels (flare-up vs. remission) with the predicted labels, and (**iv**) a heatmap illustrating the relative abundance of the most discriminative genera between the two classes (flare-up vs. remission).

**Figure 6 microorganisms-13-01941-f006:**
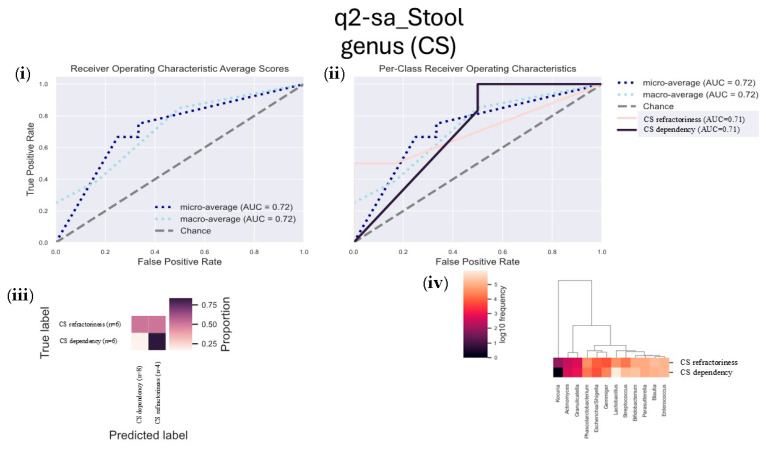
Longitudinal analysis comparing the abundance of genera in CS-dependent versus CS-refractory stool samples after 3 months of therapy. This figure compares the genus abundance in stool samples from patients who are corticosteroid (CS)-dependent and those who are CS-refractory after 3 months of treatment. The following plots are included: (**i**) the receiver operating characteristic (ROC) curve showing the overall performance of the model, (**ii**) per-class ROC curves for CS-dependent and CS-refractory groups, (**iii**) a confusion matrix comparing actual versus predicted labels for the two classes, and (**iv**) a heatmap showing the relative abundance of the most discriminating genera between CS-dependent and CS-refractory subjects. The heatmap uses “F” for flare-up subjects and “R” for remission subjects.

**Table 1 microorganisms-13-01941-t001:** Selected patients.

Code	Age	Sex	Diagnosis Age	Disease Duration	Extension of Disease Before Therapy	Severity of Disease Before Therapy	Endoscopic Mayo Sub-Score Before Cinnora Therapy	Endoscopic Mayo Sub-Score 3-Months Cinnora Therapy	Endoscopic Activity Index of Rectum 3-Months Cinnora Therapy	Endoscopic Mayo Sub-Score 6-Months Cinnora Therapy	Disease State 6-Months Cinnora Therapy	Cinnora ^R^	CorticoSteroid	Azathioprine	5-aminosalicylic Acid Derivatives	Why Start Anti-TNFa	Comorbidities	CRP	ESR	Fecal Calprotectin ng/mL	Platelets	WBC	Hb	MCV	Vitamin D	Serum Iron	Ferritin	Alcohol Habits	Smoking Habits
**101**	40	F	31	9	Left-Sided Colitis	Moderate to Severe	3	2	R	3	F-up	+	+	+	+	CS-d	No	12	38	103.32	411,000	9200	11.6	92	19	45	318	No	No
**102**																				72.63									
**103**																				146.04									
**201**	21	M	21	0.5	Pancolitis	Moderate	3	3	F-up	-	-	+	−	+	+	CS-r	No	11	19	132.77	420,000	7500	14	91	18	ND	ND	No	No
**202**																				103.4									
**203**																				−									
**301**	38	M	26	11	Left-Sided Colitis	Moderate to Severe	3	3	R	3	F-up	+	+	+	+	CS-r	RA	55	80	142.72	507,000	11,700	10.1	77	ND	ND	72	Yes	Yes
**302**																				138.9									
**303**																				132.77									
**401**	30	F	24	5	Left-Sided Colitis	Moderate	3	0	R	0	R	+	−	+	+	CS-r	No	ND	5	22.22	25,800	7800	13.8	ND	38	ND	20	No	No
**402**																				21.21									
**403**																				0									
**501**	23	F	19	4	Pancolitis	Severe	3	1	F-up	0	R	+	−	+	+	CS-r	No	ND	ND	119.41	ND	ND	ND	ND	ND	ND	ND	Yes	Yes
**502**																				0									
**503**																				0									
**601**	34	M	34	0.7	Left-Sided Colitis	Severe	3	1	R	3	F-up	+	+	+	+	CS-d	No	8	24	111.12	38,000	12,000	13.2	85	14.5	ND	ND	No	Yes
**602**																				0									
**603**																				120.49									
**701**	35	F	30	5	Pancolitis	Severe	3	2	R	1	R	+	−	−	+	CS-d	No	9	21	110.95	55,000	110,000	10.8	ND	ND	ND	ND	No	No
**702**																				133.18									
**703**																				84.41									
**801**	38	F	30	8	Pancolitis	Severe	3	2	R	1	R	+	−	+	+	CS-r	No	7	28	113.52	250,000	5400	11.9	ND	18	ND	9	No	No
**802**																				96.52									
**803**																				−									
**901**	18	M	16	2.5	Left-Sided Colitis	Severe	3	1	R	0	R	+	+	+	+	CS-r	No	ND	ND	77.69	60,000	1600	4.9	ND	54	140	65	Yes	Yes
**902**																				0									
**903**																				0									
**1001**	30	M	30	0.5	Left-Sided Colitis	Mild to Moderate	3	0	R	0	R	+	−	+	+	No CS	No	2+	17	39.43	28,100	10.5	13.1	88	ND	109	123	No	No
**1002**																				14.91									
**1003**																				0									
**1101**	26	F	22	5	Pancolitis	Severe	3	1	R	1	R	+	−	+	+	CS-d	PSC	12	57	128.29	48,600	8700	ND	ND	21	9	2	No	No
**1102**																				102.67									
**1103**																				90.99									
**1201**	32	M	29	3	Left-Sided Colitis	Moderate to Severe	3	3	F-up	ND	ND	+	+	+	+	CS-d	No	ND	ND	99.17	375,000	6200	11.8	78	ND	ND	ND	No	No
**1202**																				152.75									
**1203**																				−									
**1301**	19	M	19	1	Pancolitis	Severe	3	0	R	0	R	+	+	+	+	CS-d	No	3	10	72.14	573,000	8730	10.1	ND	ND	ND	ND	Yes	Yes
**1302**																				−									
**1303**																				0									
**1401**	64	F	59	5	Left-Sided Colitis	Moderate to Severe	3	ND	ND	1	R	+	−	−	-	CS-d	No	ND	ND	137.58	ND	ND	ND	ND	ND	ND	ND	No	No
**1402**																				−									
**1403**																				143.46									
**1501**	35	M	35	0.5	Pancolitis	Severe	3	2	R	3	F-up	+	+	+	+	CS-r	No	11	25	152.42	248,000	8900	13.8	ND	12	ND	50	No	No
**1502**																				149.1									
**1503**																				−									
**1601**	31	F	24	6	Pancolitis	Severe	3	2	R	1	R	+	+	−	+	CS-r	PG	ND	ND	145.21	ND	ND	ND	ND	ND	ND	ND	No	No
**1602**																				117.01									
**1603**																				0									
**1701**	38	M	24	13	Pancolitis	Severe	3	1	R	0	R	+	+	+	+	CS-r	No	ND	ND	148.03	239,000	13,600	15.8	ND	ND	ND	ND	No	No
**1702**																				97.43									
**1703**																				0									
**1801**	41	F	36	5	Pancolitis	Severe	3	1	R	3	F-up	+	+	−	+	CS-r	No	ND	ND	149.93	ND	ND	ND	ND	ND	ND	ND	No	No
**1802**																				0									
**1803**																				−									
**1901**	36	F	20	16	Pancolitis	Severe	3	0	R	0	R	+	+	+	-	CS-r	No	47	25	121.9	500,000	8400	14	ND	ND	ND	ND	No	No
**1902**																				0									
**1903**																				0									
**2001**	26	F	23	2	Pancolitis	Severe	3	0	R	0	R	+	−	−	+	CS-r	No	0	60	109.79	313,000	6600	8.9	ND	ND	ND	ND	No	No
**2002**																				22.29									
**2003**																				6.86									
**2101**	24	M	14	10	Pancolitis	Severe	3	2	R	2	R	+	−	+	+	CS-d	PSC	ND	ND	133.35	ND	ND	ND	ND	ND	ND	ND	No	Yes
**2102**																				140.56									
**2103**																				128.62									
**2201**	37	F	35	2	Left-Sided Colitis	Severe	3	2	R	3	F-up	+	−	−	+	CS-d	No	ND	ND	129.03	28,800	78,000	12.7	ND	12	ND	170	No	No
**2202**																				151.43									
**2203**																				−									
**2301**	22	M	19	3	Pancolitis	Severe	3	1	R	ND	ND	+	−	+	+	CS-d	PG	10	20	139.86	237,000	4900	13.3	ND	10	54	45	No	No
**2302**																				7.20									
**2303**																				−									

All patients were selected following ulcerative colitis (UC) flare-up (F-up) and were either dependent on or refractory to corticosteroid therapy. Corticosteroid dependency (CS-d) and corticosteroid refractoriness (CS-r) were the reasons for initiating treatment with Cinnora^®^, an anti-TNF alpha biologic drug, as part of UC management. “R” denotes remission of the disease, determined by the Endoscopic Mayo sub-score after 3 or 6 months of Cinnora^®^ therapy. Comorbidities observed alongside UC included primary sclerosing cholangitis (PSC), rheumatoid arthritis (RA), and pyoderma gangrenosum (PG). Various clinical variables were assessed, including C-reactive protein (CRP), erythrocyte sedimentation rate (ESR), white blood cell count (WBC), hemoglobin (Hb), and mean corpuscular volume (MCV). Cases where tests were not performed are indicated as “ND”.

## Data Availability

The DNA sequences generated and analyzed during the current study are available in the NCBI SRA data_BioProject ID: PRJNA930373 “Inflamed gut-mucosa biopsies and fecal microbiomes in Ulcerative colitis (UC) patients treated with anti-TNFalpha (Cinnora^®^)”.

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
