# Peer review of "Predictive Microbial Markers Distinguish Responders and Non-Responders to Adalimumab: A Step Toward Precision Medicine in Ulcerative Colitis"

_microorganisms, 2025, doi:10.3390/microorganisms13081941_

Round 1
Reviewer 1 Report
Comments and Suggestions for Authors
This manuscript conducted a prospective longitudinal analysis of 23 patients with moderate-to-severe ulcerative colitis (UC), revealing that adalimumab responders exhibited increased Bacteroidetes and decreased Proteobacteria in their fecal microbiota, along with enrichment of beneficial bacteria such as Faecalibacterium prausnitzii. Conversely, mucosal microbiota demonstrated persistent dysbiosis. The research highlights niche-specific microbiota changes, with fecal microbiota recovery correlating with clinical remission, while mucosal dysbiosis may reflect subclinical inflammation, providing microbial biomarkers for precision UC therapy. The rigorous study design (longitudinal multi-timepoint sampling + dual-niche analysis) offers valuable microbial markers for personalized UC treatment. However, several limitations should be noted:
1.The small sample size (23 patients total: 14 responders vs. 9 non-responders), particularly the inadequate non-responder cohort, may compromise statistical power (e.g., non-significant α-diversity differences). Statistical power calculations (e.g., via G*Power) should be supplemented or conclusions explicitly framed as exploratory.
2.The significant increase in Proteobacteria (especially γ-Proteobacteria) in responder mucosal microbiota (Fig. 1, Table 3) contradicts known associations of this phylum with disease exacerbation (e.g., Escherichia coli-mediated pro-inflammatory effects). While authors attribute this to "subclinical inflammation," no supporting data on immune microenvironment or host-microbiota interactions (e.g., cytokine profiles, barrier gene expression) were provided. This warrants additional discussion or acknowledgment as a limitation.
3.PICRUSt2-inferred metabolic pathways (e.g., fimbriae/biofilm-related genes) lack experimental validation (e.g., metagenomics/metabolomics). This indirect evidence should be emphasized, with explicit discussion of functional inference limitations.
4.Missing clinical data marked as "ND" (not detected) in Table 1 require clarification of handling methods (e.g., inclusion/exclusion in statistical models).
5.β-diversity analysis should be supplemented with PCA or PCoA visualization results.
6.The use of species' Latin names needs standardization. The full name should appear upon first mention, followed by the abbreviated genus name (italicized throughout). The Latin names of 292-296 should be corrected.
Author Response
Comments and Suggestions for Authors
This manuscript conducted a prospective longitudinal analysis of 23 patients with moderate-to-severe ulcerative colitis (UC), revealing that adalimumab responders exhibited increased Bacteroidetes and decreased Proteobacteria in their fecal microbiota, along with enrichment of beneficial bacteria such as Faecalibacterium prausnitzii. Conversely, mucosal microbiota demonstrated persistent dysbiosis. The research highlights niche-specific microbiota changes, with fecal microbiota recovery correlating with clinical remission, while mucosal dysbiosis may reflect subclinical inflammation, providing microbial biomarkers for precision UC therapy. The rigorous study design (longitudinal multi-timepoint sampling + dual-niche analysis) offers valuable microbial markers for personalized UC treatment. However, several limitations should be noted:
- The small sample size (23 patients total: 14 responders vs. 9 non-responders), particularly the inadequate non-responder cohort, may compromise statistical power (e.g., non-significant α-diversity differences). Statistical power calculations (e.g., via G*Power) should be supplemented or conclusions explicitly framed as exploratory.
- R1: The number of enrolled patients (n=23) was determined based on the available budget, technical resources, and laboratory capacity at the time of the study. Unfortunately, the study was conducted during the COVID-19 pandemic, which posed significant logistical and clinical challenges, limiting further recruitment and sampling. Nevertheless, we believe that our longitudinal and dual-niche design provides meaningful insights, and we are planning to expand the sample size in future follow-up studies.
- Inclusion and Exclusion Criteria
Patients were eligible for inclusion if they met all of the following conditions:
- A confirmed diagnosis of ulcerative colitis (UC) based on colonoscopy and histopathological evaluation by a gastrointestinal pathologist.
- Clinical assessment by a gastroenterology specialist confirming moderate-to-severe UC and inadequate response to conventional treatment with 5-aminosalicylic acid (5-ASA) compounds.
- Decision to initiate anti-TNFα therapy (adalimumab) as part of routine clinical management.
Patients were excluded if they met any of the following conditions:
- A diagnosis of any other autoimmune disease or inflammatory condition unrelated to UC.
History of gastrointestinal cancer or any other malignancy at the time of enrollment.
Unfortunately, the total number of subjects, as well as the number per group, represents a limitation of this analysis, and a formal power analysis is not required to recognize this constraint. Nevertheless, we performed a power estimation using the R packages lme4 and broom.mixed (rather than G*Power, which is overly simplistic for data of this complexity), considering both ecosystems separately (stool and tissue) and combined. By setting a minimum acceptable statistical power of 50%—which we acknowledge is relatively low—we found some taxa to be significant in the stool analysis and in the combined stool and tissue sample analysis. Specifically, Bacteroides vulgatus and Bacteroidaceae were significant in stool samples, while Clostridiales Family XIII. Incertae Sedis and Bacillales incertae sedis emerged from the combined analysis. Based on these results, we will revise the manuscript to emphasize that these findings must be interpreted in the context of the limited sample size. On the other hand, it is also important to highlight the budgetary constraints of this project, which prevented the inclusion of more subjects, as well as the fact that the study was conducted during the COVID-19 pandemic, which further delayed the timeline required to achieve these results.
Considering all statements, we added a sentence at the start of the discussion regarding the limitations of this study. “Although the total number of subjects, as well as the number per group, represents a limitation of this analysis, we believe that our longitudinal and dual-niche design provides meaningful insights, and we are planning to expand the sample size in future follow-up studies.”
- The significant increase in Proteobacteria (especially γ-Proteobacteria) in responder mucosal microbiota (Fig. 1, Table 3) contradicts known associations of this phylum with disease exacerbation (e.g., Escherichia coli-mediated pro-inflammatory effects). While authors attribute this to "subclinical inflammation," no supporting data on immune microenvironment or host-microbiota interactions (e.g., cytokine profiles, barrier gene expression) were provided. This warrants additional discussion or acknowledgment as a limitation.
- R2: We chose to revise the text in 'Asymptomatic inflammation,' considering that the stool microbiome showed improvement in the initial dysbiosis level, and an improvement of the clinical condition, although the mucosal microbial flora exhibited signs of deterioration and inflammation along the treatment.
- PICRUSt2-inferred metabolic pathways (e.g., fimbriae/biofilm-related genes) lack experimental validation (e.g., metagenomics/metabolomics). This indirect evidence should be emphasized, with explicit discussion of functional inference limitations.
- R3: To explore the potential functional alterations in the gut microbiome of UC patients undergoing Adalimumab therapy, predicted metabolic pathways were inferred using PICRUSt2 (Phylogenetic Investigation of Communities by Reconstruction of Unobserved States 2). This approach enabled the estimation of microbial gene content from 16S rRNA gene sequencing data based on known databases such as KEGG and MetaCyc [24-26]. Previously published data in newly diagnosed Type 1 Diabetes patients showed that a good correlation existed between the pathways found with a proteomic approach and the ones inferred with PICRUSt based on 16S sequencing indeed 8 of the 13 pathways previously identified through a direct metaproteomics approach [27] were also detected among the dysregulated pathways predicted by the PICRUSt computational pipeline using our set of data [28].
- Missing clinical data marked as "ND" (not detected) in Table 1 require clarification of handling methods (e.g., inclusion/exclusion in statistical models).
- R4: done. We revised Table 1 and added missing Fecal Calprotectine values
- β-diversity analysis should be supplemented with PCA or PCoA visualization results.
- R5: done. “PERMANOVA analysis revealed no statistically significant differences among Pre-treated, Responsive, and Non-Responsive samples treated with the TNF-alpha inhibitor Cinnora®, either collectively or in pairwise comparisons (Supplementary Figure S3).”
- The use of species' Latin names needs standardization. The full name should appear upon first mention, followed by the abbreviated genus name (italicized throughout). The Latin names of 292-296 should be corrected.
- R6: done

Reviewer 2 Report
Comments and Suggestions for Authors
Despite being a single cohort, the differences between Non-responders vs responders is an important finding. This major finding need to be highlight in the title. The current title it just doesn't sell well.
Onerous tables do not help the reader see the longitudinal changes. Stackable bar graphs (at the very least) and/or progressive bubble graphs could help the presentation enormously.
Discussion needs to include the findings on Key Pathways Enriched in Mucosal Biopsies but Not Stool Microbiomes .
Author Response
Reviewer 2:
Comments and Suggestions for Authors
Despite being a single cohort, the differences between Non-responders vs responders is an important finding. This major finding need to be highlight in the title. The current title it just doesn't sell well.
R – We modify in: “Predictive Microbial Markers Distinguish Responders and Non-Responders to Adalimumab: A Step Toward Precision Medicine in Ulcerative Colitis.”
Onerous tables do not help the reader see the longitudinal changes. Stackable bar graphs (at the very least) and/or progressive bubble graphs could help the presentation enormously.
R – We modify Table 2 and Table 3 in Figure 3 and Figure 4 using the differential abundance analysis computed with the EdgeR algorithm. We decided to maintain the Supplementary tables, because richer in informations, since they were computed with 3 different algorithms.
Discussion needs to include the findings on Key Pathways Enriched in Mucosal Biopsies but Not Stool Microbiomes .
R – We added the analysis on Stools and we inserted this text regarding it: “Using PICRUSt2 analysis of 16S sequencing data from stool microbiomes, 347 KEGG-inferred pathways were statistically significant. Among these, 48 pathways showed increased abundance after six months of TNF-α inhibitor treatment. Specifically, seven pathways were related to energy metabolism, six to cell wall/envelope biogenesis, five to membrane transport and drug resistance, four each to carbohydrate/amino acid metabolism, and to stress responses. Finally, three are associated with protein folding and DNA repair, and two with Transporters and Channels. Among the 251 inferred pathways that showed a decrease in the abundances after 6 months of TNF-α inhibitor administration, the great majority (162) were pathways associated with Metabolic pathways, and sugar and Amino acid metabolism.”

Round 2
Reviewer 2 Report
Comments and Suggestions for Authors
My previous comments have been satisfactorily addressed.